# Possible Synergistic Antidiabetic Effects of Quantified *Artemisia judaica* Extract and Glyburide in Streptozotocin-Induced Diabetic Rats via Restoration of PPAR-α mRNA Expression

**DOI:** 10.3390/biology10080796

**Published:** 2021-08-18

**Authors:** Abdulaziz S. Saeedan, Gamal A. Soliman, Rehab F. Abdel-Rahman, Reham M. Abd-Elsalam, Hanan A. Ogaly, Khalid M. Alharthy, Maged S. Abdel-Kader

**Affiliations:** 1Department of Pharmacology, College of Pharmacy, Prince Sattam Bin Abdulaziz University, Al-Kharj 11942, Saudi Arabia; a.binsaeedan@psau.edu.sa (A.S.S.); g.soliman@psau.edu.sa (G.A.S.); k.alharthy@psau.edu.sa (K.M.A.); 2Department of Pharmacology, Faculty of Veterinary Medicine, Cairo University, Giza 12211, Egypt; 3National Research Centre, Department of Pharmacology, Giza 12622, Egypt; rf.abdelrahman@nrc.sci.eg; 4Department of Pathology, Faculty of Veterinary Medicine, Cairo University, Giza 12211, Egypt; rehammahmoudpathology@gmail.com; 5Department of Chemistry, College of Science, King Khalid University, Abha 61421, Saudi Arabia; ohanan@kku.edu.sa; 6Department of Biochemistry, Faculty of Veterinary Medicine, Cairo University, Giza 12211, Egypt; 7Department of Pharmacognosy, College of Pharmacy, Prince Sattam Bin Abdulaziz University, Al-Kharj 11942, Saudi Arabia; 8Department of Pharmacognosy, College of Pharmacy, Alexandria University, Alexandria 21215, Egypt

**Keywords:** *Artemisia judaica*, streptozotocin, glyburide, PPAR-α, rats

## Abstract

**Simple Summary:**

A considerable number of diabetic patients are in favour of using oral antidiabetic drugs in combination with certain herbs instead of using oral antidiabetic drugs alone. *Artemisia judaica* (AJ) is one of the herbs documented to have antidiabetic effects. This study examined the effect of using combination of *A. judaica* extract (AJE) and the oral hypoglycemic drug glyburide (GLB, 5 mg/kg) on diabetic rats. Fasting blood glucose (FBG), insulin levels, glycated hemoglobin (HbA1c) percentage, serum lipid profile, and oxidative stress biomarkers were estimated. The histopathological examination of the pancreas and the immunohistochemical analysis of anti-insulin, anti-glucagon, and anti-somatostatin protein expressions were also performed. The analysis of the hepatic mRNA expression of peroxisome proliferator-activated receptor α (PPAR-α) and nuclear factor erythroid 2-related factor-2 (Nrf2) genes was performed using quantitative reverse transcription-polymerase chain reaction (qRT-PCR). Combination of GLB and 500 mg/kg of AJE highly improved FBG, insulin levels, HbA1c, and lipid profile in blood when compared with GLB monotherapy. Furthermore, GLB plus 500 mg/kg of AJE combination was the most successful in restoring insulin content in the *β*-cells and diminished the levels of glucagon and somatostatin of the α- and δ-endocrine cells in the pancreatic islets, restoring PPAR-α and Nrf2 mRNA expression in the liver. In conclusion, these data indicate that GLB plus 500 mg/kg of AJE combination gives greater glycemic improvement than GLB monotherapy.

**Abstract:**

Several members of the genus *Artemisia* are used in both Western and African traditional medicine for the control of diabetes. A considerable number of diabetic patients switch to using oral antidiabetic drugs in combination with certain herbs instead of using oral antidiabetic drugs alone. This study examined the effect of *Artemisia judaica* extract (AJE) on the antidiabetic activity of glyburide (GLB) in streptozotocin (STZ)-induced diabetes. Forty-two male Wistar rats were divided into seven equal groups. Normal rats of the first group were treated with the vehicle. The diabetic rats in the second–fifth groups received vehicle, GLB (5 mg/kg), AJE low dose (250 mg/kg), and AJE high dose (500 mg/kg), respectively. Groups sixth–seventh were treated with combinations of GLB plus the lower dose of AJE and GLB plus the higher dose of AJE, respectively. All administrations were done orally for eight weeks. Fasting blood glucose (FBG) and insulin levels, glycated hemoglobin (HbA1c) percentage, serum lipid profile, and biomarkers of oxidative stress were estimated. The histopathological examination of the pancreas and the immunohistochemical analysis of anti-insulin, anti-glucagon, and anti-somatostatin protein expressions were also performed. The analysis of the hepatic mRNA expression of PPAR-α and Nrf2 genes were performed using quantitative RT-PCR. All treatments significantly lowered FBG levels when compared with the STZ-control group with the highest percentage reduction exhibited by the GLB plus AJE high dose combination. This combination highly improved insulin levels, HbA1c, and lipid profile in blood of diabetic rats compared to GLB monotherapy. In addition, all medicaments restored insulin content in the β-cells and diminished the levels of glucagon and somatostatin of the α- and δ-endocrine cells in the pancreatic islets. Furthermore, the GLB plus AJE high dose combination was the most successful in restoring PPAR-α and Nrf2 mRNA expression in the liver. In conclusion, these data indicate that the GLB plus AJE high dose combination gives greater glycemic improvement in male Wistar rats than GLB monotherapy.

## 1. Introduction

Diabetes is a metabolic disorder multifaceted by impaired carbohydrates, fat, and protein metabolism due to the lack of insulin secretion and/or increased tissue resistance to insulin. Based on the World Health Organization (WHO) statistics, the number of global diabetic patients is expected to approach 366 million by 2030 [1]. Protocols for the treatment of diabetes depend on the use of insulin and other oral hypoglycemic drugs such as biguanides, sulphonylureas, α-glycosidase inhibitors, and amylin analogues, which, at higher doses, result in adverse effects ranging from diarrhea, lactic acidosis, liver problems, and hypoglycemia [2]. Many herbal medicines are also recommended for the management of high glucose blood level. It is declared that up to 72.8% of diabetic people use herbal remedies for their effectiveness, fewer adverse effects, and the relative lower cost [3]. Many of the medicinal plants are thought to have significant antidiabetic benefits and have been used in the treatment of diabetes [4,5]. The leaves of the olive plant (*Olea europaea* L.) have been used for centuries in folk medicine to manage diabetes [6]. *Momordica charantia*, also referred to as bitter melon, is commonly used as a traditional treatment for diabetes in Asia, Africa, and South America [7].

Members of the genus *Artemisia* belonging to the family Asteraceae are generally small herbs and/or shrubs with more than 480 species [8]. Many of *Artemisia* species have been well studied concerning the treatment of diabetes [9,10,11] and several members of the genus are used in both Western and African traditional medicine for the control of diabetes [12]. Moreover, some species of the genus possess diverse biological activities such as anti-inflammatory, antioxidant, antimalarial, antibacterial, and antiseptic [13]. AJ (Shih Baladi) is an aromatic plant widely distributed in Sinai Peninsula, Jordan, Palestine, and Saudi Arabia [14]. It is used in the Arabian Gulf folk medicine to manage numerous disorders, including diabetes, and for treating parasites infestation [15,16,17]. Further, Jordanian Bedouins are using plant infusion for the control of both diabetes and sexual dysfunction [18]. The hypoglycemic effect of AJE was traced to the eudesmanolid, vulgarin, and its isomer epivulgarin [19].

Several patients with diabetes are recognized to use medicinal plants with antidiabetic characteristics in addition to oral antidiabetic drugs. Concurrent use of these plants and oral antidiabetic drugs may lead to interactions with each other, resulting in drug–herb interactions. For instance, St. John’s wort extract was found to inhibit the Cytochrome P450 family 2 subfamily C member 9 (CYP2C9) and Cytochrome P450 family 3 subfamily A member 4 (CYP3A4), and the conventional antidiabetics such as glibenclamide and rosiglitazone are substrates of CYP2C9, whereas pioglitazone and repaglinide are substrates of CYP3A4. Therefore, diabetic patients receiving these combinations should be carefully monitored for possible signs of reduced efficacy [20]. Another example of possible herb–drug interaction is the increased efficacy when using Karela or *Momordica charantia* fruit extract together with metformin or glibenclamide, allowing for reduced doses of metformin [21]. In this investigation, the outcomes of using GLB plus AJE were evaluated and compared with the antidiabetic efficacy of GLB monotherapy.

## 2. Materials and Methods

### 2.1. Plant Material and Extraction

The total ethanol extract of authenticated samples of AJ was quantified for the active sesquiterpenes vulgarin and epivulgarin as described earlier [22,23] and used in the current study.

### 2.2. Experimental Animals

Healthy adult male albino Wistar rats (250 ± 10 g), in-house bred at the Animal House Colony of the National Research Centre (NRC), Egypt were procured for the study. All animals were kept in polypropylene cages under standard environmental conditions (temperature 25 ± 2 °C, relative humidity 55 ± 10% and 12:12 light:dark cycle) and food and water were freely accessible to them. The experimental protocol complied with the National Institutes of Health Guide lines and was approved by the Institutional Animal Care and Use Committee, Cairo University (approval no.: CU-II-F-86-18).

### 2.3. Induction of Experimental Diabetes

Rats were fasted overnight before intraperitoneal injection of STZ dissolved in 0.1 M citrate buffer (pH 4.5) (Sigma-Aldrich Corp, St. Louis, MO, USA) at a dose of 60 mg/kg body weight to induce diabetes [24]. Control rats were administered only citrate buffer. Rats with blood glucose level >250 mg/dL measured by Accu-Chek Performa blood glucose meter (Roche Diagnostic, Germany) after three days from STZ injection in blood samples collected from the tail vein by were assigned as diabetic and included in the study.

### 2.4. Experimental Design

Diabetic and non-diabetic rats were randomly arranged into seven groups (*n* = 6) and received the following treatments: Group I: Non-diabetic control (NC) + the vehicle (3% Tween 80), Group II: STZ control + the vehicle, Group III: STZ + GLB (5 mg/kg), Group IV: STZ + AJE at 250 mg/kg (AJE-250), Group V: STZ + AJE at 500 mg/kg (AJE-500), Group VI: STZ + GLB (5 mg/kg) + AJE-250, and Group VII: STZ + GLB (5 mg/kg) + AJE-500. GLB and AJE were administered as suspension in 3% Tween 80. GLB and AJE were given orally, once daily using an oral tube for eight successive weeks. The animals’ body weights were recorded at the start of the study (0-time), at the end of weeks 2, 4, and 8.

### 2.5. Estimation of Biochemical Parameters

Blood samples were collected into sampling tubes through retro-orbital venous plexus from pentobarbital sodium (35 mg/kg, ip) anesthetized rats fasted overnight at week 0 and the ends of the 2nd, 4th, and 8th weeks of treatment. Blood samples were centrifuged for 20 min at 5000 rpm to separate the sera.

#### 2.5.1. Effect on Blood Glucose and Insulin Levels

Both FBG and insulin levels in serum were estimated in the collected samples using the commercially available Spinreact ELISA kits (Spain) and Cobas ELISA kits (Belgium) following the manufacturer’s manual, respectively.

#### 2.5.2. Effect on Total Hemoglobin and Glycosylated Hemoglobin Levels

Other blood samples were similarly collected from each animal at the end of the experiment (after 8 weeks of the medication period) into tubes containing ethylenediamine tetraacetic acid (EDTA) as an anticoagulant for the assessment of both total hemoglobin (Hb) and glycosylated hemoglobin (HbA1c) with the aid of commercially available kits (QCA, Spain).

#### 2.5.3. Effect on Blood Lipid Profile

The blood samples obtained at the end of the 8th week of treatments were used to assess the lipid profile. Levels of triglycerides (TGs), total cholesterol (TC), and high density lipoprotein cholesterol (HDL-C) were determined in serum spectrophotometrically using commercially available kits. Low density lipoprotein (LDL-C) concentrations were estimated according to the following formula given by Friedewald et al. [25] as follows: LDL-C = TC − [HDL-C + TG/5)] where TG/5 is equivalent to the amount of very low-density lipoprotein (VLDL)-cholesterol.

### 2.6. Tissue Collection

At the end of the experiment and after blood collection, rats were euthanized via cervical decapitation. Liver, kidney, and pancreas tissue samples were removed, carefully washed in ice-cold saline, and stored at −80 °C until the time of analysis.

### 2.7. Effect on Oxidative Stress and Lipid Peroxidation (LPO) Markers

Pancreas tissues in 0.1 M Tris-HCl (pH 7.4) were separately homogenized and then centrifuged for 10 min at 1700 rpm. The obtained supernatants were preserved at −80 °C and used for biochemical assessments. The levels of innate antioxidant enzymes as glutathione peroxidase (GPx), superoxide dismutase (SOD), catalase (CAT), as well as the reduced glutathione (GSH) and malondialdehyde (MDA) levels were determined using Biodiagnostic assay kits (Egypt) following the manufacturer’s instructions.

### 2.8. Histopathological Investigation of Pancreas

Pieces of pancreatic tissues from all groups were separately collected and fixed in 10% neutral buffered formalin for 24 h and used for obtaining 3–4 µm paraffin embedding sections, following the methods described by Abdel-Rahman et al. [26].

### 2.9. Immunohistochemical Analysis

The immunohistochemical investigation of the insulin expression in the pancreatic islets of the different experimental groups was done according to the methods described by Abdel-Rahman et al. [27] and Khamis et al. [28]. Firstly, the tissue sections were deparaffinized, rehydrated, and antigenically retrieved by methods described by Abu-Elala et al. [29]. Secondly, tissue sections were incubated with mouse monoclonal insulin (Sc-8033; Santa Cruz Biotechnology, Inc.; Dallas, TX, USA) at a dilution of 1:100, mouse monoclonal anti-glucagon antibody (MABN238; Millipore) at a dilution of 1:8000, and rat monoclonal anti-somatostatin antibody (MAB354; Millipore) at a dilution of 1:100 for overnight, followed by adding the blocking solution to block the endogenous peroxidase activity. The tissue sections were incubated with a sheep anti-mouse antibody (AQ300D; Millipore) and goat anti-rat antibody (AP136P; Millipore) for 10 min; then, sections were incubated with streptavidin peroxidase (Thermo Fisher Scientific; Waltham, MA, USA). At the end, tissue sections were incubated with 3,3′-diaminobenzidine tetrahydrochloride (DAB; Sigma) for 10 min to visualize the reaction. In each field, the immune-positive areas were analyzed by using image analysis software (Image J, version 1.46a, NIH, Bethesda, MD, USA) in 7 microscopic high-power fields (X400). Calculation of the percentage of positive stained area (%) was done. The morphometric analysis of the pancreatic islet cells composition was accomplished to estimate the percent of insulin positive β-cells/total islets area, as well as α-cell/total islet area % and δ-cell/total islet area %, according to the approach stated by Abdel-Rahman et al. [24].

### 2.10. Quantitative Reverse Transcription-Polymerase Chain Reaction (qRT-PCR) Analysis

Total RNA was extracted from frozen liver tissue samples using a TRIzol RNA Isolation Reagent (Invitrogen) and then quantified by a NanoDrop 2000 spectrophotometer. Real-time RT-PCR reactions were carried out by using RNA samples templates, and the condition used for the RT-PCR reaction was: 50 °C for 5 min for cDNA synthesis, 95 °C for 2 min, then 40 cycles of 95 °C for 15 s, 55 °C for 15 s and 72 °C for 20 s. The amplification curves were specific primers for PPAR-α and Nrf2 genes, and One-Step SYBR GreenER Kit (Invitrogen). The thermal analyzed by a software (QIAGEN) to obtain the Ct values of target genes and β-actin (reference gene). The relative mRNA expression of each gene was calculated as fold change of the negative control after normalization to β-actin expression [30]. The primer sequences are presented in Table 1.

### 2.11. Statistical Analysis

Data are presented as mean ± SEM. The obtained data were statistically evaluated by one-way ANOVA as well as Tukey’s multiple comparison post hoc test using GraphPad Prism; version 5.0 (GraphPad Software, Inc., San Diego, CA, USA). Data were considered statistically significant when *p* ≤ 0.05.

## 3. Results

### 3.1. Effect on Body Weight

Changes in the body weights of the control and experimental rats are displayed in Table 2. A significant reduction in the body weights of STZ-control animals was observed until the end of the study (8 weeks’ treatment) when compared with treated groups. By the end of the study period, 12.02% reduction in body weight was recorded in the diabetic untreated group. GLB, AJE-250, AJE-500, and GLB plus AJE-250 reversed the STZ-mediated body weight reduction after 2, 4, and 8 weeks of medications. At the end of 8 weeks of treatment, GLB caused an increase in body weight by 28.34%. Similarly, AJE-250, AJE-500, and GLB plus AJ-250 caused increases in body weight by 29.12%, 34.25%, and 32.73% respectively. AJ-250, AJE-500, and GLB plus AJE-250 did not result in any change in body weight in comparison with the GLB-treated group. The combination of GLB and AJE-500 exhibited a superior effect on the percentage of weight gain of STZ-control animals at the end of the 2nd until the end of the 8th week of treatment. At these times, the combination increased the body weights of diabetic rats by 11.45%, 29.14%, and 44.64%, respectively. Interestingly, GLB plus AJE-500 combination therapy normalized the body weight of STZ-diabetic animals after 4 weeks of treatment.

### 3.2. Effect on Blood Glucose Levels

The levels of FBG in the control and experimental groups are depicted in Table 3. At baseline, the vehicle and other medication groups were matched with respect to the FBG level. However, STZ-control rats showed increased FBG concentrations when compared to NC rats, which remained significantly (*p* ≤ 0.05) elevated until the end of the experiment. Following 2, 4, and 8 weeks of GLB treatment, diabetic rats showed significantly (*p* ≤ 0.05) reduced FBG levels (186.2 ± 6.53, 169.2 ± 8.62, and 135.3 ± 7.43 mg/dL, respectively) in comparison with STZ-control rats (352.2 ± 9.58, 337.8 ± 16.47, and 326.5 ± 16.52 mg/dL, respectively). The antidiabetic effects of AJE-250, AJE-500, and GLB plus AJE-250 were comparable to that of GLB. Maximum reductions in FBG level of AJE-250 (57.81%), AJE-500 (65.93%), and GLB plus AJE-250 (63.37%) treated groups were observed on week 8. Importantly, GLB plus AJE-500 treatment induced more statistically significant reduction in the FBG level of diabetic rats when compared with those exposed to GLB alone. Furthermore, GLB plus AJE-500 allowed to restore the FBG levels in diabetic rats to normal values (112.6 ± 8.62 and 96.62 ± 6.52 mg/dL) after 4 and 8 weeks of treatment, respectively.

### 3.3. Effect on Blood Insulin Levels

As noted in Table 4, the serum levels of insulin in STZ-control group at the end of the second, fourth, and eighth week of the medication period were significantly reduced (3.4 ± 0.10, 3.6 ± 0.28, and 3.5 ± 0.27 U/L, respectively), compared with the NC group (7.3 ± 0.48, 7.4 ± 0.57, and 7.2 ± 0.48 U/L, respectively). However, treatment of diabetic rats with GLB for 2, 4, and 8 weeks significantly increased serum insulin levels (3.9 ± 0.11, 4.7 ± 0.32 and 5.2 ± 0.36 U/L, respectively) in comparison to the STZ-control group. Additionally, there were no significant variations in the serum level of insulin between diabetic animals treated with AJE-250, AJE-500, and GLB + AJE-250 and those treated with GLB monotherapy. The STZ-diabetic group treated with GLB and AJE-250 together did not reach statistical significance change in insulin level to enable a comparison with the GLB-treated group. However, a statistically significant variation was noticed in favor of the combination of GLB and AJE-500 by the end of weeks 2, 4, and 8 of treatment. The obtained results showed the highest serum insulin level in the STZ-diabetic group which received the combination of GLB and AJE-500 after 4 and 8 weeks of medication. At these periods, the blood insulin levels of the GLB plus AJE-500 treated group (5.9 ± 0.29 and 6.5 ± 0.41 U/L, respectively) are comparable with those of NC rats (7.4 ± 0.57 and 7.2 ± 0.48 U/L, respectively).

### 3.4. Effect on Total Hemoglobin and Glycosylated Hemoglobin Levels

Table 5 depicts the levels of total hemoglobin and the percentages of HbA1c in blood of different groups of animals after 8 weeks of the medication period. The level of total hemoglobin was markedly decreased, while difference percentage of HbA1c was significantly increased in STZ-control rats in comparison to difference NC group. Administration of GLB, AJE-250, AJE-500, and GLB plus AJE-250 blocked the above alterations and significantly (*p* ≤ 0.05) improved the level of total hemoglobin and the percentages of HbA1c towards normal levels. The concomitant administration of GLB and AJE-500 exhibited significant improvement in the blood level of Hb and percentage of HbA1c, compared to the GLB-treated group, and almost normalized their values.

### 3.5. Effect on Serum Lipid Profile

Serum lipid profiles in different groups after 8 weeks of the medication period are described in Table 6. In difference STZ-control group, there were significant elevations in TGs (49.3 ± 2.55 mg/dL) and TC (66.2 ± 1.32 mg/dL) compared to the non-diabetic group (27.7 ± 0.97 mg/dL and 46.5 ± 1.27 mg/dL, respectively). Administration of AJE (250 and 500 mg/kg) to diabetic rats significantly decreased their serum levels of TGs and TC, compared to the STZ-control group. Furthermore, the GLB plus AJE-250 combination significantly decreased TGs and TC in diabetic rats. The effects of AJ extracts and GLB plus AJE-250 combination were comparable to those of GLB monotherapy. On the other hand, serum levels of HDL-C were significantly decreased in difference STZ-control group in comparison with the NC group. Administration of GLB, AJE-250, and AJE-500 markedly elevated the serum level of HDL-C, in comparison with the STZ-control group. The STZ-control rats exhibited marked elevations in LDL-C and VLDL-C levels, in comparison with NC rats. The diabetic rats treated with GLB, AJE-250, and AJE-500 showed significantly decreased LDL-C and VLDL-C levels, in comparison with the STZ-control group. Furthermore, marked ameliorations were noticed in the lipid profile in the diabetic group exposed to the combination of GLB and AJE-500. Administration of the GLB and AJE-500 combination to diabetic animals tended to improve TG, TC, HDL-C, LDL-C, and VLDL-C levels to their normal values.

### 3.6. Effect on Oxidative Stress and Lipid Peroxidation Markers

Table 7 explains the impact of GLB and AJE on the oxidative stress and LPO biomarkers in the pancreatic tissues of control and experimental groups. The pancreatic tissues of the STZ-control group displayed a significant decrease in SOD, GPx, CAT, and GSH contents along with an elevation in MDA content when compared against the NC group. Upon oral dosing of GLB, AJE-250, AJE-500, or GLB plus AJE, there were marked elevations in the levels of SOD, GPx, CAT, and GSH in pancreatic homogenates of animals in comparison with STZ-control rats. Additionally, MDA levels were significantly (*p* ≤ 0.05) reduced in response to these treatments as compared against the STZ-control group. Interestingly, the GLB and AJE-500 combination restored the activities of the antioxidant enzymes and the MDA contents in pancreatic homogenates of diabetic rats compared with GLB-treated rats, and nearly normalized their levels.

### 3.7. Histopathological Investigation of Pancreas

The NC group showed an almost normal pancreatic architecture, as the islets of Langerhans appeared with a central core of β-cells and peripheral mantle of α- and δ-cells (Figure 1a). However, the STZ treated group revealed a massive reduction in the number of β-cells of the islets of Langerhans with the appearance of apoptotic and necrosed cells. Papillary hyperplasia of the epithelial lining and severe dilatation of pancreatic duct were also recorded. The STZ + GLB- and STZ + AJE-250-treated groups showed moderate apoptosis and necrosis of β-cells (Figure 1c,d). The group treated with both STZ + AJE-500 and STZ + GLB + AJE-250 showed moderate improvement in the number of β-cells with less deterioration in the texture of the islets of Langerhans (Figure 1e,f). The groups treated with STZ + GLB + AJE-500 revealed a marked hypertrophy of the islets of Langerhans with an elevation in the β-cell number with the maintenance of the islet morphology. The STZ + GLB + AJE-500 treated group was considered the group that was improved the most (Figure 1g).

### 3.8. Immunohistochemical Analysis of Insulin, Glucagon, and Somatostatin

Figure 2, Figure 3 and Figure 4 summarize the results of the content of pancreatic islets for insulin, glucagon, and somatostatin in the different treated groups. The NC group revealed a strongly immune-positive insulin reaction which was located in β-cells in most of the pancreatic islets (Figure 2a). Glucagon immune-positive reaction was observed in *α*-cells that were found in the peripheral area of the pancreatic islet (Figure 3a). Somatostatin immune-positive reaction was localized in δ-cells forming an incomplete circle in the pancreatic islets (Figure 4a). The STZ-control group showed a significant reduction in the insulin content of β-cells (Figure 2b), β-cell/total islet area compared to the NC group, a significant elevation of both glucagon and somatostatin contents in pancreatic islets (Figure 3b and Figure 4b), as well as α-cell/total islet area % and δ-cell/total islet area %. On the other hand, the groups treated with GLB, AJE-250, AJE-500, GLB + AJE-250, and GLB + AJE-500 showed a significant elevation in the insulin contents of the β-cells, β-cell/total islet area (Figure 2c–h), a significant reduction of glucagon, somatostatin contents, α-cell/total islet area % and δ-cell/total islet area % in comparison to the STZ-control group (Figure 3 and Figure 4c–h). No significance difference was observed between the GLB + AJE-500 group and the NC group.

### 3.9. Real Time-PCR for Hepatic Gene Expression Analysis

The current work was executed to recognize how GLB and AJE treatments, as either monotherapy or combined therapy, could regulate hepatic expression of the two transcriptional factors PPARα and Nrf-2 and to explore the possible regulatory link between the two factors to mediate the antidiabetic, antisteatotic, and antioxidant action of the suggested treatments. Based on the RT-PCR results, a decreased expressional level of PPARα was observed in the liver tissue of STZ-control rats as compared with the NC group. Following AJE administration at 250 and 500 mg/kg and GLB at 5 mg/kg to rats for 8 weeks, hepatic PPARα mRNA expression was significantly augmented when compared to the STZ-control group (Figure 5A). Furthermore, a downregulation of Nrf-2 was shown in the liver tissues of STZ-control rats compared to that in the NC rats. In contrast, Nrf-2 was overexpressed by the anti-oxidative effect of two doses of AJE (250 and 500 mg/kg), also in the GLB-treated group compared to the STZ-control group. Remarkably, a combination of AJE (250 and 500 mg/kg) with GLB exhibited a more efficient anti-oxidative action demonstrated by significant upregulation of gene expression of Nrf-2 compared to diabetic animals and the values in monotherapy groups (Figure 5B).

## 4. Discussion

Diabetes was induced in experimental animals following intraperitoneal injection of STZ (60 mg/kg) to monitor the possible drug–herb interaction by difference combined administration of GLB and AJE in STZ-diabetic rats in comparison with GLB monotherapy.

STZ-induced diabetes is usually accompanied by an intense reduction in body weight [31]. The body weight reduction in diabetic animals might be a result of tissue protein degeneration and muscle wasting [32]. In our investigation, STZ-control rats gained less body weight all over the experimental period as compared to non-diabetic animals. Importantly, the beneficial impact of the GLB plus AJE-500 therapy on body weight of diabetic animals was more effectual than the GLB monotherapy. This combination therapy improved the weight of diabetic animals to values that are similar to those of NC animals. The efficiency of the combination of GLB and AJE-500 to conserve body weight in diabetic animals might be explained by the high ability to control hyperglycemia.

The blood concentrations of glucose and insulin express the glycemic state of diabetic patients. In the present investigation, the STZ-control group revealed high concentration of blood glucose and low level of serum insulin in comparison to the normal animals. Administration of GLB to diabetic rats lowered the FBG and elevated the serum level of insulin compared with STZ-control animals. However, FBG and insulin levels have not returned to normal values. The GLB hypoglycemic effect is produced via the stimulation of β-cells to release insulin and the suppression of glucagon secretion [33]. Hence, the presence of a considerable mass of β-cells able to secrete insulin is necessary for GLB to act. The insulin level in the group of diabetic rats indicated that some β-cells are intact and able to synthesize and secrete insulin. FBG and serum insulin levels were comparable among the GLB, AJE-250, AJE-500, and GLB plus AJE-250 groups. Furthermore, the obtained results indicated the additive effect of the combination between GLB and AJE-500 since the levels of FBG and insulin in diabetic rats were not normalized by the single treatment of GLB or AJE-500 alone. The combination of GLB and AJE-500 was the most efficacious in reducing the elevated blood glucose and restoring the insulin levels all over the experimental period. Administration of the GLB and AJE-500 combination resulted in stable levels of FBG and serum insulin within the normal physiological ranges. Interestingly, the advantageous impact on glycemic control observed for the GLB plus AJE-500 combination occurred without any observed increase danger of hypoglycemia.

Several studies have reported that the hypoglycemic effects of *Artemisia* plants were comparable with those of the standard antidiabetic medications repaglinide, insulin, metformin, and GLB [2,34,35,36]. Confirming the outcomes of previous investigations, the treatment of diabetic animals with some plants of *Artemisia* species (*A. sieberi, A. pallens* and *A. judaica*) induced a marked decrease in FBG [15,37,38]. The antidiabetic variations between GLB and AJE may be due to the presence of active components in the extract. Phytochemical screening of the AJE yielded flavonoids, saponins, terpenes, and tannins [39]. Flavonoids inhibit cAMP phosphodiesterase, which is a modulator of insulin secretion [15]. Furthermore, Nazaruk and Borzym-Kluczyk [40] mentioned that terpenoids from *A. turanica* exerted an antidiabetic effect via the improvement of insulin release from the β-cells as well as lowering of the cellular resistance to insulin. Different mechanisms of action have been mentioned in the literature to describe the potential effect of *Artemisia* plants as antidiabetics. Aggarwal et al. [41] proposed that the antidiabetic mechanism of *Artemisia* plants might be due to improving the function of β-cells and restoring pancreatic islets. Another study considered the improvement of the carbohydrate metabolism dysfunction associated with diabetes as another possible mechanism for antidiabetic action of some *Artemisia* plants [42]. Interestingly, Bhat et al. [17] reported that AJE modulates serum glucose levels by inhibiting the key blood sugar modulating enzymes, namely: α-amylase, α-glucosidase and dipeptidyl peptidase IV. Vulgarin from AJE reported to have oral hypoglycemic effect [19]. Furthermore, the significant anti-hyperglycemic activity of AJE may be due to the existence of the two isomers of thujone and represents about 3.2% of the essential oil collectively [43]. Thujone can increase the free insulin-stimulated glucose transporter by activation of the adenosine monophosphate-activated protein kinase [36].

Glycated hemoglobin (HbA1c) is considered as a distinguished marker of glycemic condition of diabetic patients. STZ-control rats showed marked reduction in Hb level and significant increase in HbA1c percentage as a reflection of poor glycemic control. The rate of Hb glycosylation gives an indication about the blood glucose level. GLB, AJE-250, AJE-500, and the combination of GLB and AJE-250 markedly decreased HbA1c percentages in comparison with the STZ-control group. Furthermore, the Hb level and HbA1c percentage were successfully controlled in animals treated with the GLB and AJE-500 combination over that recorded in the GLB-treated group. By the end of the study, both Hb level and HbA1c percentage were returned to normal values following GLB plus AJE-500 administration. GLB, AJE, and their combinations versus Hb glycation were, in diminution order, of GLB + AJE-500 > AJE-500 > GLB + AJE-250 > GLB > AJE-250. This beneficial result reflects the effectiveness in controlling diabetes by the GLB and AJE-500 combination. Additionally, the results suggest that the combination might be efficacious for the long-term management of diabetes.

Diabetes complications include abnormalities in lipid metabolism manifested by increasing levels of blood TC, LDL-C, and VLDL-C as well as decreased HDL-C [44]. In this respect, Bhowmik et al. [45] demonstrated that diabetic dyslipidemia comprises a triad of elevated LDL-C/HDL-C ratio in addition to hypertriglyceridemia. Normally, TGs are hydrolyzed by lipoprotein lipase enzyme (LLE) that is stimulated by insulin. However, in case of diabetes, LLE is not stimulated due to an insulin insufficiency that results in increased synthesis of TGs by the liver and a disproportion in the liberation and rate of clearance of VLDL-C by LLE [46]. Thus, TGs are usually used as indicators of intracellular aggregation of lipids [47]. Consistent with this, the results of the present study showed significantly higher levels of TGs, TC, and VLDL-C and reduced HDL-C levels in the STZ-control group compared to non-diabetic rats. The elevated level of TGs in the STZ–control group observed in the present investigation may be due to a shortage of insulin [48]. Furthermore, dosing of GLB, AJE-250, and AJE-500 to diabetic rats produced significant reductions in their serum levels of TGs, TC, and VLDL-C, as well as elevated HDL-C levels. The ameliorative effect of GLB, AJE-250, and AJE-500 on the blood lipoprotein profile of diabetic rats may be due to a rise in insulin release. The reduced TC and elevated HDL-C levels after AJE treatment are remarkable, as it has been stated that the majority of drugs used in the management of hypercholesterolaemia decrease both TC and HDL-C levels [49]. In concurrence with other studies, plants of *Artemisia* species exhibited a noticeable hypolipidemic effect and inverted the lipid profile changes in diabetic animals [50,51]. The antihyperlipidemic activity of AJE may be attributed to vulgarin, flavonoids, and polyphenols via the stimulation of pancreatic insulin secretion, augmentation of glucose oxidation, and increase in the lipid synthesis pathway [51]. Interestingly, the combination of GLB and AJE-500 had a preferable impact on regulating serum lipids than GLB. This combination was able to reduce the levels of TGs, TC, LDL-C, and VLDL-C, but elevated the levels of HDL-C in diabetic rats to normal levels that may be indicative of the powerful antidiabetic activity induced by the combination of GLB and AJE-500. This effect might be attributed to an increased secretion of pancreatic insulin that stimulates fatty acid synthesis and accumulation of fatty acids into hepatic TG and adipose tissue.

In this study, we also examined the role of GLB, AJE, and their combination in the protection of diabetic rats against oxidative stress. Oxidative stress induced by excess reactive oxygen species (ROS) and decreased antioxidant capacity is considered a key agent in the progression of diabetes [52]. ROS are under strict control of the endogenous antioxidant defense mechanisms that include both enzymatic and non-enzymatic pathways. Enzymatic antioxidants include superoxide dismutase (SOD), glutathione peroxidase (GPx), and catalase (CAT) enzymes. GSH is a non-enzymatic antioxidant found in most forms of aerobic life and plays an important function in keeping cellular antioxidant capacity. Some reports have proposed that oxidative stress is a usual pathogenic agent for the dysfunction of the pancreatic β-cells [53]. Pancreatic β-cells are especially susceptible to ROS, because they are low in antioxidant enzymes [54,55]. Thus, the capability of the oxidative stress to injure mitochondria and suppress insulin release is not surprising [56].

In the present study, the reduced activities of the antioxidant enzymes in the pancreatic tissues of STZ-control animals indicate that their pancreatic tissues were under oxidative stress. Further, reduced tissue content of GSH has been considered as a marker of oxidative stress [57]. The present results demonstrated that the pancreatic content of GSH was significantly reduced in the STZ-control group. However, GLB, AJE-250, and AJE-500 medications markedly elevated SOD, GPx, CAT, and GSH accompanied by reduced MDA contents in the pancreatic tissue, demonstrating their capability to protect against oxidative stress in diabetic animals. The significant role of GLB and AJE against decreased antioxidant enzymes activities and GSH depletion in STZ-diabetic animals may be attributed to their antihyperglycemic effect.

The protective effect of GLB plus AJE-500 against the pancreatic oxidative stress of diabetic animals was higher than that recorded in rats that received GLB alone. The combination enhanced the pancreatic content of the antioxidant enzymes and GSH in diabetic animals to near-normal values. Further, administration of GLB plus AJE-500 for 8 weeks has markedly normalized the disturbed pancreatic content of MDA in diabetic animals. These effects indicate that the GLB plus AJE-500 combination has a synergistic activity against oxidative stress and LPO in pancreatic tissues of STZ-control animals. Restoring the levels of the antioxidant enzymes and MDA in the pancreatic homogenates may protect β-cells against ROS and LPO.

Since the GLB plus AJE-500 combination induced further improvement in the antioxidant activity than that induced by GLB or AJE alone, it is proposed that AJE may be acting by a different mechanism than that of GLB. Phytochemical analysis of AJ demonstrated that it is a large source of total polyphenols [17]. Other reports have mentioned a potent correlation between total phenolic contents of the plants and their antioxidant activities [58]. Accordingly, the antioxidant activity of AJ may be attributed to its total polyphenol content.

Accumulating scientific evidence reported that the abnormal accumulation of TG in the diabetic liver is due to the simultaneous activation of lipogenesis and gluconeogenesis that lead to excessive lipid production [45]. PPARα, a transcriptional factor predominantly expressed in the liver, plays key role in maintaining lipid homeostasis through regulation of various enzymes in the lipid and glucose metabolism [59]. It has been reported that PPARα expression is downregulated under diabetic stress in both human and animal models [60,61]. Activation of PPARα has been shown to improve lipid and glucose metabolism in diabetes by reducing hyperglycemia and increasing insulin sensitivity. Hence, PPARα activators could alleviate liver injuries during diabetic pathogenesis and other metabolic dysfunction associated disorders [60,62].

To further elucidate the mechanism underlying the observed antihyperlipidemic effect of AJE, GLB, and their combination, the current study examined the hepatic expression levels of PPARα. In line with previous studies, we reported that PPARα expression level decreased in the liver tissue of STZ-control diabetic animals [60]. Interestingly, the current study demonstrates that AJE upregulated the mRNA expression of PPARα in the liver of diabetic rats (Figure 5A), suggesting that AJE directly increased PPARα transcriptional activity. The improving effects of PPARα on lipid metabolism may explain our finding that PPARα activation modulated the STZ-induced dyslipidemia.

Activation of β-oxidation of fatty acids is mediated thru PPARα, and further has the prospect to trigger redox-sensitive pathways involved in cyto-defense such as Nrf2 pathway. Therefore, the crosstalk between PPARα and Nrf2 has been proposed to act as a vital role in regulating the stress response [63]. The findings of the present study support this previous evidence. Nrf2 acts as a central regulator of the cellular redox potential by regulating the transcription of various endogenous antioxidant and detoxify enzymes [64,65]. In normal cells, Nrf2 is kept in the cytoplasm via its binding to Kelch-like ECH-associated protein 1 (Keap1) which also contributes to Nrf2 degradation by ubiquitination. Under stressed conditions, the Keap1 system is disrupted by the excessively produced ROS. This disruption allows Nrf2 translocation to the nucleus, in which Nrf2 binds antioxidant response elements (AREs) in the promoter region of several antioxidant genes, initiating their transcription [66]. Several studies have showed that uncontrolled diabetes significantly reduced Nrf2 expression levels in different tissues, where activation of the Nrf2 signaling pathway prevents the development of diabetes and diabetic complications [67,68,69]. In line with previous reports, the obtained results revealed that Nrf2 transcriptional level was downregulated in the hepatic tissues of rats in response to STZ-induced diabetes (Figure 5B). AJE administration effectively prevented Nrf2 downregulation in the liver compared to STZ-control. We previously reported that AJE exhibited a protective effect against diabetes-induced testicular dysfunction by activation of the Nrf2/HO-1 pathway [23]. Upregulation of Nrf2 and its downstream target antioxidant genes could contribute to the observed improved liver redox status in AJE and GLB treated groups.

There are some limitations in the present study. One of the limitations is that we used the Wistar rat of a certain age and sex. Different strains and model organisms such as non-human primates, mice, or others may elicit different effectiveness or response profile. Although the beneficial effect of vulgarin [19] and thujone [36] to lower blood glucose level were reported, in the current study, their interaction with GLB was not explored. Further phytochemical studies are in progress to isolate, characterize the active compounds, and evaluate their possible interaction with oral hypoglycemic drugs.

## 5. Conclusions

In conclusion, the objective of the study was fulfilled as the herb–drug interactions were found to be evident and most significant in male diabetic Wistar rats exposed to a GLB plus AJE-500 combination. Moreover, the present investigation showed that the pathway mediating the synergy between AJE and GLB includes the up-regulation of liver PPARα and Nrf2 expression. The results suggest that AJE is one of the therapeutic options for the medication of diabetic patients who have already undergone GLB therapy. In addition, the dose or frequency of GLB has to be altered when it is concomitantly administered with AJE, in order to avoid any unexpected serious acute hypoglycemic shock, as GLB and AJE were reported to lower blood glucose through different mechanisms.

## Figures and Tables

**Figure 1 biology-10-00796-f001:**
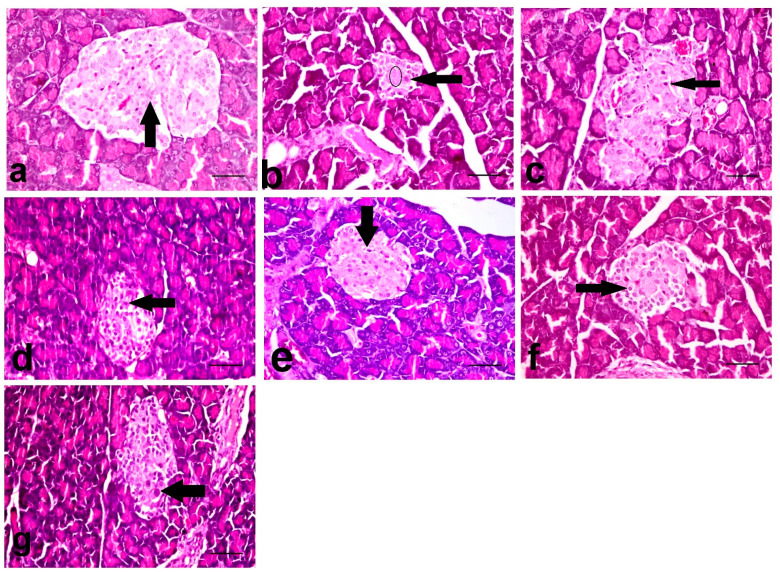
The histopathological photomicrographs of the pancreases. (**a**) NC group showing normal β-cells (arrow) (**b**) STZ-control group showing marked decrease in the number (circle), necrosis (arrow) and apoptosis of β-cells, and distortion of the islet. (**c**) STZ + GLB and (**d**) STZ + AJE-250 showing increase in the number of β-cells with necrosis (arrow) and vacuolation of some cells. (**e**) STZ + AJE-500 and (**f**) STZ + GLB + AJE-250 showing moderate improvement of the β-cells with individual cell necrosis (arrow). (**g**) STZ + GLB + AJE-500 group showing single cell necrosis of β-cells (arrow); scale bar, 25 μm.

**Figure 2 biology-10-00796-f002:**
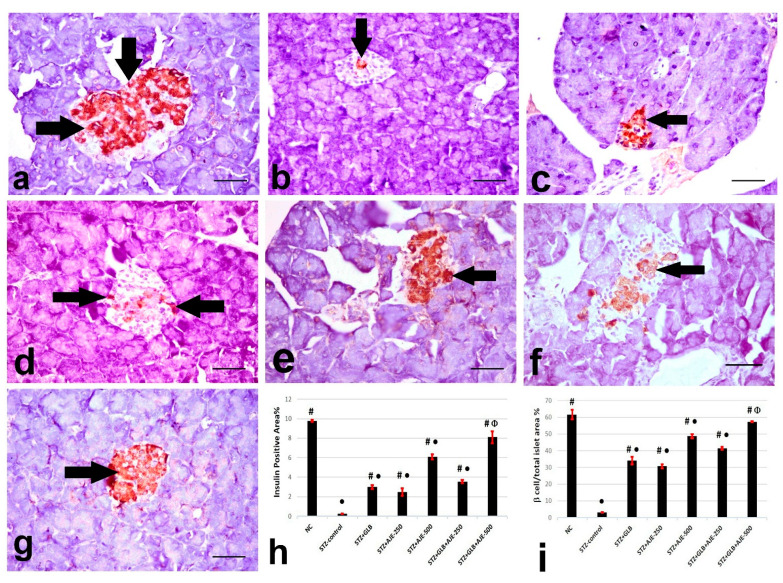
Representative insulin immunohistochemistry in β-cells (arrows) of the pancreatic islets of the different experimental groups: (**a**) NC group, (**b**) STZ-control group, (**c**) STZ + GLB-treated group, (**d**) STZ + AJE-250, (**e**) STZ + AJE-500 group, (**f**) STZ + GLB + AJE-250, and (**g**) STZ + GLB + AJE-500 group. Scale bar, 25 μm. (**h**) Bar chart represents the insulin content of β-cells %. (**i**) β-cell/total islet area %. Data are presented as the mean ± SEM (*n* = 7). ● Statistically significant difference from the NC group at *p* ≤ 0.05. # Statistically significant difference from the STZ-control group at *p* ≤ 0.05. Փ Statistically significant difference from the GLB group at *p* ≤ 0.05.

**Figure 3 biology-10-00796-f003:**
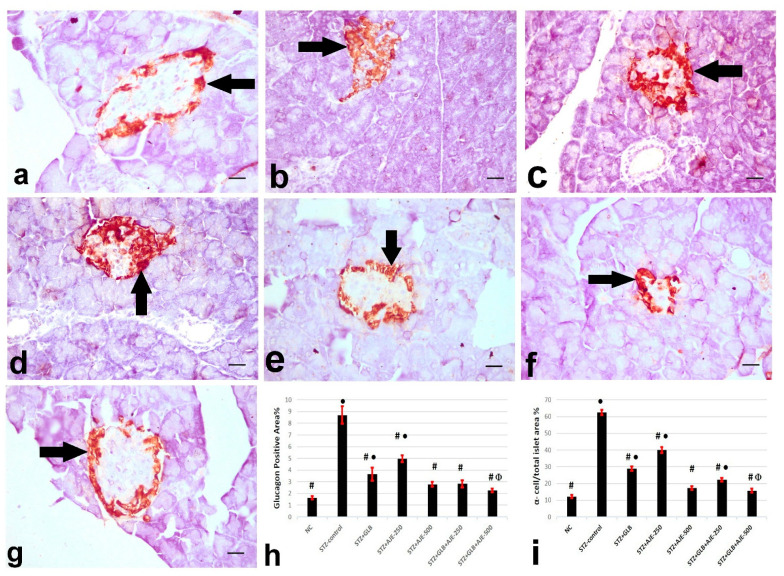
Represented glucagon immunohistochemistry in α-cells (arrows) of the pancreatic islets of the different experimental groups: (**a**) NC group, (**b**) STZ-control group, (**c**) STZ + GLB-treated group, (**d**) STZ + AJE-250, (**e**) STZ + AJE-500 group, (**f**) STZ + GLB + AJE-250, and (**g**) STZ + GLB + AJE-500 group; scale bar, 25 μm. (**h**) Bar chart represents the glucagon content of α-cells %. (**i**) α- cell/total islet area %. Data are presented as the mean ± SEM (*n* = 7). ● Statistically significant difference from the NC group at *p* ≤ 0.05. # Statistically significant difference from the STZ-control group at *p* ≤ 0.05. Փ Statistically significant difference from the GLB group at *p* ≤ 0.05.

**Figure 4 biology-10-00796-f004:**
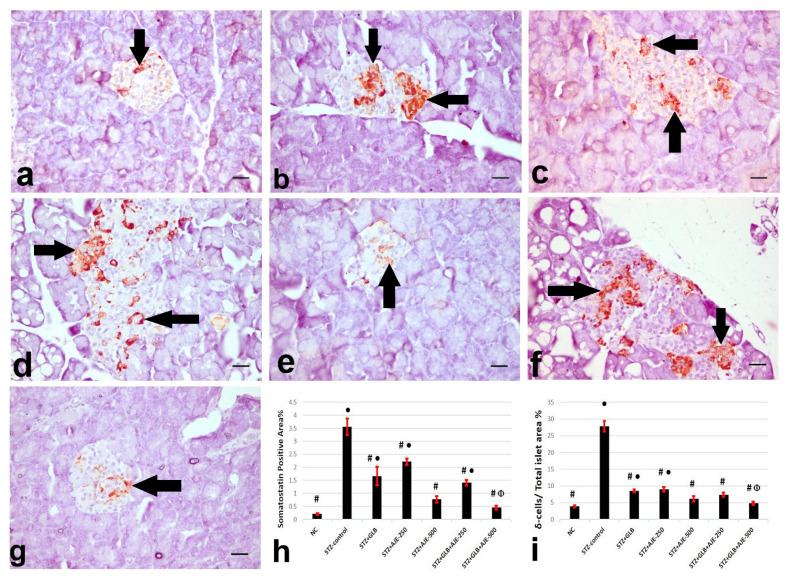
Represented somatostatin immunohistochemistry in the δ-cells (arrows) of the pancreatic islets of the different experimental groups. (**a**) NC group, (**b**) STZ-control group, (**c**) STZ + GLB-treated group, (**d**) STZ + AJE-250, (**e**) STZ + AJE-500 group, (**f**) STZ + GLB + AJE-250, and (**g**) STZ + GLB + AJE-500 group; scale bar, 25 μm. (**h**) Bar chart represents the glucagon content of δ-cells %. (**i**) δ-cell/total islet area %. Data are presented as the mean ± SEM (*n* = 7). ● Statistically significant difference from the NC group at *p* ≤ 0.05. # Statistically significant difference from the STZ-control group at *p* ≤ 0.05. Փ Statistically significant difference from the GLB group at *p* ≤ 0.05.

**Figure 5 biology-10-00796-f005:**
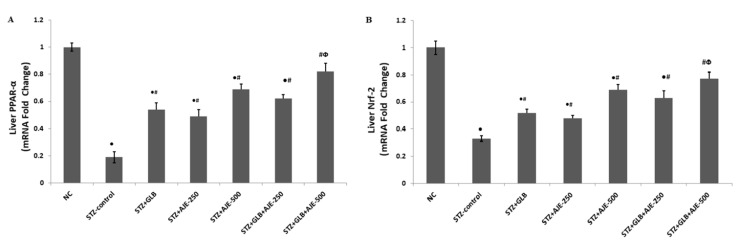
Effect of GLB, AJE, and their combination on hepatic mRNA expression of PPARα (**A**) and Nrf-2 (**B**) in STZ-diabetic groups. Data presented as mean ± SEM (*n* = 6) relative to the mRNA level in the NC and after being normalized to β-actin mRNA level. Multiple group comparisons were performed by analysis of variance (ANOVA) followed by Tukey’s multiple comparison post hoc test at *p* ≤ 0.05. ● Statistically significant difference from the NC group at *p* ≤ 0.05. # Statistically significant difference from the STZ-control group at *p* ≤ 0.05. Փ Statistically significant difference from GLB group at *p* ≤ 0.05.

**Table 1 biology-10-00796-t001:** Oligonucleotide primer sequences.

Gene	Primer Sequence (5′-3′)		Accession no.
PPAR-α	Forward	TTCGGAAACTGCAGACCT	NM_013196.1
	Reverse	TTAGGAACTCTCGGGTGAT	
Nrf2	Forward	CACATCCAGACAGACACCAGT	XM_006234398.3
	Reverse	CTACAAATGGGAATGTCTCTGC	
β actin	Forward	ATGGTGGGTATGGGTCAG	NM_031144.3
	Reverse	CAATGCCGTGTTCAATGG	

**Table 2 biology-10-00796-t002:** Effect of GLB, AJE, and their combination on body weights of STZ-diabetic rats.

Treatment Groups	% of Body Weight Gain (g)		
	2 Weeks	4 Weeks	8 Weeks
NC	13.65 ± 0.88	33.96 ± 1.82	49.84 ± 2.93
STZ-control	−2.90 ± 0.18 ●	−7.15 ± 0.31 ●	−12.02 ± 0.47 ●
STZ + GLB	7.16 ± 0.41 ●#	14.71 ± 0.96 ●#	28.34 ± 2.11 ●#
STZ + AJE-250	7.35 ± 0.43 ●#	15.72 ± 0.97 ●#	29.12 ± 1.98 ●#
STZ + AJE-500	8.52 ± 0.75 ●#	18.17 ± 1.74 ●#	34.25 ± 1.75 ●#
STZ + GLB + AJE-250	7.86 ± 0.55 ●#	17.51 ± 1.47 ●#	32.73 ± 1.95 ●#
STZ + GLB + AJE-500	9.45 ± 0.76 ●#Փ	29.14 ± 1.73 #Փ	44.64 ± 2.82 #Փ

Data presented as mean ± SEM (*n* = 6). ● Statistically significant difference from the NC group at *p* ≤ 0.05. # Statistically significant difference from the STZ-control group at *p* ≤ 0.05. Փ Statistically significant difference from GLB group at *p* ≤ 0.05. Multiple group comparisons were done by analysis of variance (ANOVA) and Tukey’s multiple comparison post hoc tests.

**Table 3 biology-10-00796-t003:** Effect of GLB, AJE, and their combination on serum levels of FBG in STZ-diabetic rats.

Treatment Groups	FBG (mg/dL)						
	0-Time	2 Weeks		4 Weeks		8 Weeks	
	M ± SEM	M ± SEM	%	M ± SEM	%	M ± SEM	%
NC	96.3 ± 5.43	94.5 ± 4.67	−1.87	95.7 ± 6.97	−0.62	94.4 ± 5.70	−1.97
STZ-control	349.2 ± 7.57 ●	352.2 ± 9.58 ●	0.86	347.8 ± 16.47 ●	−0.40	340.5 ± 16.52 ●	−2.49
STZ + GLB	347.1 ± 5.08 ●	186.2 ± 6.53 ●#	−46.36	169.2 ± 8.62 ●#	−51.25	135.3 ± 7.43 ●#	−61.02
STZ + AJE-250	339.4 ± 9.87 ●	194.7 ± 9.20 ●#	−42.63	175.3 ± 8.50 ●#	−48.35	143.2 ± 8.11 ●#	−57.81
STZ + AJE-500	347.8 ± 8.97 ●	176.5 ± 8.18 ●#	−49.25	152.1 ± 7.1 7●#	−56.27	118.5 ± 7.92 ●#	−65.93
STZ + GLB + AJE-250	345.9 ± 7.66 ●	179.6 ± 8.22 ●#	−48.08	161.3 ± 8.50 ●#	−53.37	126.7 ± 6.37 ●#	−63.37
STZ + GLB + AJE-500	358.5 ± 6.98 ●	157.5 ± 6.49 ●#Փ	−56.07	112.6 ± 8.62 #Փ	−68.59	96.62 ± 6.52 #Փ	−73.05

Data presented as mean ± SEM (*n* = 6). ● Statistically significant difference from the NC group at *p* ≤ 0.05. # Statistically significant difference from the STZ-control group at *p* ≤ 0.05. Փ Statistically significant difference from GLB group at *p* ≤ 0.05. Multiple group comparisons were done by analysis of variance (ANOVA) and Tukey’s multiple comparison post hoc tests.

**Table 4 biology-10-00796-t004:** Effect of GLB, AJE, and their combination on serum levels of insulin in STZ-diabetic rats.

Treatment Groups	Insulin (U/L)						
	0-Time	2 Weeks		4 Weeks		8 Weeks	
	M ± SEM	M ± SEM	%	M ± SEM	%	M ± SEM	%
NC	7.5 ± 0.34	7.3 ± 0.48	−2.67	7.4 ± 0.57	−1.33	7.2 ± 0.48	−4.00
STZ-control	3.8 ± 0.26 ●	3.4 ± 0.10 ●	−10.53	3.6 ± 0.28 ●	−5.26	3.5 ± 0.27 ●	−7.89
STZ + GLB	3.5 ± 0.18 ●	3.9 ± 0.11 ●#	11.43	4.7 ± 0.32 ●#	34.29	5.2 ± 0.36 ●#	48.57
STZ + AJE-250	3.5 ± 0.25 ●	3.8 ± 0.10 ●#	8.57	4.5 ± 0.25 ●#	28.57	5.0 ± 0.30 ●#	42.86
STZ + AJE-500	3.4 ± 0.20 ●	4.2 ± 0.28 ●#	23.53	5.3 ± 0.32 ●#	55.88	5.9 ± 0.33 ●#	73.53
STZ + GLB + AJE-250	3.6 ± 0.17 ●	4.0 ± 0.23 ●#	21.21	4.8 ± 0.31 ●#	45.45	5.3 ± 0.39 ●#	60.61
STZ + GLB + AJE-500	3.5 ± 0.21 ●	4.6 ± 0.25 ●#Փ	31.43	5.9 ± 0.29 #Փ	68.57	6.5 ± 0.41 #Փ	85.71

Data presented as mean ± SEM (*n* = 6). ● Statistically significant difference from the NC group at *p* ≤ 0.05. # Statistically significant difference from the STZ-control group at *p* ≤ 0.05. Փ Statistically significant difference from GLB group at *p* ≤ 0.05. Multiple group comparisons were done by analysis of variance (ANOVA) and Tukey’s multiple comparison post hoc tests.

**Table 5 biology-10-00796-t005:** Effect of GLB, AJE, and their combination on blood levels of total Hb and HbA1c in STZ-diabetic rats after 8 weeks of the medication period.

Treatment Groups	Total Hemoglobin (mg/dL)	HbA1c (%)
NC	14.4 ± 0.64	3.9 ± 0.21
STZ-control	9.7 ± 0.36 ●	8.5 ± 0.36 ●
STZ + GLB	11.1 ± 0.48 ●#	6.3 ± 0.48 ●#
STZ + AJE-250	11.4 ± 0.57 ●#	6.7 ± 0.30 ●#
STZ + AJE-500	12.3 ± 0.62 ●#	5.0 ± 0.42 ●#
STZ + GLB + AJE-250	12.0 ± 0.60 ●#	5.5 ± 0.29 ●#
STZ + GLB + AJE-500	13.6 ± 0.65 #Փ	4.5 ± 0.32 #Փ

Data presented as mean ± SEM (*n* = 6). ● Statistically significant difference from the NC group at *p* ≤ 0.05. # Statistically significant difference from the STZ-control group at *p* ≤ 0.05. Փ Statistically significant difference from GLB group at *p* ≤ 0.05. Multiple group comparisons were done by analysis of variance (ANOVA) and Tukey’s multiple comparison post hoc tests.

**Table 6 biology-10-00796-t006:** Effect of GLB, AJE, and their combination on lipid profile in blood of STZ-diabetic rats after 8 weeks of the medication period.

Treatment Groups	TGs (mg/dL)	TC (mg/dL)	HDL-C (mg/dL)	LDL-C (mg/dL)	VLDL (mg/dL)
NC	27.7 ± 0.97	44.5 ± 2.27	24.6 ± 0.74	14.4 ± 1.15	5.5 ± 0.28
STZ-control	49.3 ± 2.55 ●	66.2 ± 1.32 ●	14.5 ± 0.75 ●	41.8 ± 1.26 ●	9.9 ± 0.45 ●
STZ + GLB	40.2 ± 1.37 ●#	59.2 ± 1.72 ●#	17.1 ± 0.63 ●#	34.1 ± 2.97 ●#	8.0 ± 0.41 ●#
STZ + AJE-250	42.4 ± 1.42 ●#	60.6 ± 2.14 ●#	16.9 ± 0.48 ●#	35.2 ± 1.58 ●#	8.5 ± 0.42 ●#
STZ + AJE-500	36.4 ± 1.55 ●#	52.6 ± 2.61 ●#	19.2 ± 0.75 ●#	26.1 ± 2.50 ●#	7.3 ± 0.36 ●#
STZ + GLB + AJE-250	37.5 ± 1.17 ●#	54.1 ± 2.27 ●#	18.9 ± 0.76 ●#	27.7 ± 1.96 ●#	7.5 ± 0.39 ●#
STZ + GLB + AJE-500	33.5 ± 2.58 #Փ	49.5 ± 1.65 #Փ	23.7 ± 0.65 #Փ	19.1 ± 1.89 #Փ	6.7 ± 0.30 #Փ

Data presented as mean ± SEM (*n* = 6). ● Statistically significant difference from the NC group at *p* ≤ 0.05. # Statistically significant difference from the STZ-control group at *p* ≤ 0.05. Փ Statistically significant difference from GLB group at *p* ≤ 0.05. Multiple group comparisons were done by analysis of variance (ANOVA) and Tukey’s multiple comparison post hoc tests.

**Table 7 biology-10-00796-t007:** Effect of GLB, AJE, and their combination on oxidative stress and LPO parameters in pancreatic tissues of STZ-diabetic rats.

Treatment Groups	SOD (U/mg protein)	GPx (U/mg protein)	CAT (U/mg protein)	GSH (µmol/g tissue)	MDA (nmol/g tissue)
NC	57.6 ± 3.20	6.8 ± 0.55	11.4 ± 0.15	9.7 ± 0.52	26.5 ± 1.76
STZ-control	21.5 ± 1.63 ●	2.2 ± 0.14 ●	5.1 ± 0.31 ●	3.8 ± 0.20 ●	47.4 ± 3.17 ●
STZ + GLB	30.6 ± 2.87 ●#	3.7 ± 0.30 ●#	7.0 ± 0.52 ●#	5.7 ± 0.47 ●#	38.2 ± 2.55 ●#
STZ + AJE-250	33.5 ± 2.23 ●#	3.8 ± 0.25 ●#	7.3 ± 0.41 ●#	6.3 ± 0.49 ●#	37.3 ± 2.27 ●#
STZ + AJE-500	40.6 ± 3.71 ●#	4.9 ± 0.47 ●#	8.3 ± 0.42 ●#	7.4 ± 0.66 ●#	32.4 ± 1.57 ●#
STZ + GLB + AJE-250	36.2 ± 2.73 ●#	4.7 ± 0.38 ●#	7.9 ± 0.55 ●#	7.0 ± 0.52 ●#	35.2 ± 2.61 ●#
STZ + GLB + AJE-500	49.5 ± 3.24 #Փ	5.5 ± 0.36 #Փ	10.2 ± 0.72 #Փ	8.5 ± 0.57 #Փ	30.7 ± 2.13 #Փ

Data presented as mean ± SEM (*n* = 6). ● Statistically significant difference from the NC group at *p* ≤ 0.05. # Statistically significant difference from the STZ-control group at *p* ≤ 0.05. Փ Statistically significant difference from GLB group at *p* ≤ 0.05. Multiple group comparisons were done by analysis of variance (ANOVA) and Tukey’s multiple comparison post hoc tests.

## Data Availability

The data presented in this study are available in the open access manuscript.

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
