# Peer review of "Possible Synergistic Antidiabetic Effects of Quantified Artemisia judaica Extract and Glyburide in Streptozotocin-Induced Diabetic Rats via Restoration of PPAR-α mRNA Expression"

_biology, 2021, doi:10.3390/biology10080796_

Round 1

Reviewer 1 Report

Please check appended pdf file.

Author Response

Section 0: Title / Abstract

*What is the purpose of a simple summary? The information within is redundant with the

abstract, the writing style is the same and I see no need to have both… Maybe a merge would be beneficial?

Simple summary is required by the journal to target non-scientific readers.

*L2/Title: By using “The possible synergistic effects” hints there are no other possibilities, whereas caution should dictate an openness to other research. My suggestion would be cutting of “The” and rewrite the title only as “Possible synergistic effects (…)”

Changed as suggested by the esteem reviewer.

*L20 / L39: “switch to using oral antidiabetic”, but switch from what?

Switch to using oral antidiabetic drugs in combination with certain herbs instead of using oral antidiabetic drugs alone.

*L22: antidiabetic effects.

Corrected as suggested by the esteem reviewer.

*LL27 / L50: analysis, not analyze.

Corrected as suggested by the esteem reviewer.

*LL30-34: there is a repetition of the idea, please reformulate the sentences to read with more

clarity.

Changed as suggested by the esteem reviewer.

*L42: Wistar, not wister.

Corrected as suggested by the esteem reviewer.

*L46: it would be good to state the route of administration: oral, subcutaneous, intraperitoneal,  etc.

Changed as suggested by the esteem reviewer.

*L51: there is no need to write PPAR-a and Nrf-2 in full as they are already explained in simple summary.

Changed as suggested by the esteem reviewer.

*LL58/59: your conclusion is supported for the wistar rat of a certain age and sex. Different strains and model organisms (such as non-human primates, mice or others may elicit a different effectiveness or response profile).

We added male Wistar rats to conclusion

Section 1: introduction

*Overall this section needs an overhaul as each paragraph comprises too many different sentences that could be merged and rewritten to improve the reading flow. The discourse is

excessively broken.

The introduction was improved as directed by the esteem reviewer.

*L65: It would be beneficial to state the geographical area to which these numbers refer to, either the USA, Europe, worldwide…

Changed as suggested by the esteem reviewer.

*L66: depend.

Corrected as suggested by the esteem reviewer.

*LL68-70: this sentence could be integrated with the previous one to improve the reading flow.

Changed as suggested by the esteem reviewer.

*L72: fewer instead of lesser

Changed as suggested by the esteem reviewer.

*LL73-75: if you describe many medicinal plants, please give at least two examples and expand to two or three references.

We added 2 examples and 2 references

*L79: please be more formal and do not use “a lot”. Many, plural, multiple, diverse, extensive, several…

Changed as suggested by the esteem reviewer.

*L82: “members of the genus” makes no sense, please rephrase.

Changed as suggested by the esteem reviewer.

*LL86-88: this sentence repeats the previous, please merge. Moreover, “sexual dysfunction” is socially more acceptable than incompetence.

Changed as suggested by the esteem reviewer.

*LL89-93: what is the connection between these phytochemical studies and the one in the manuscript? As it sits, seems only as random information with no direct link to the present findings.

This part was removed with the cited reference.

Section 2: Results

*Results description should be more linear, example, by group / treatment, instead of going back and forth.

Changed as suggested by the esteem reviewer.

*As methods section comes at the end of the manuscript, abbreviations not used at earlier times should be written in full, such as the case of serum lipid profile components and oxidative stress / peroxidation markers.

The section of material and methods was moved after the introduction and the abbreviations were explained in full when used for the first time. In addition, we added a paragraph for the abbreviations.

*Please use the same formatting for the graphics and figures as the remaining text to have some coherence.

Changed as suggested by the esteem reviewer.

*L109: there is no amelioration of body weight. There may be a reversion of a trend of weight gain/loss, changes can be averted/prevented/precluded but an amelioration does not make sense in a physiological parameter as it implies a worsening and should be applied (for example) to pathologies.

Changed as suggested by the esteem reviewer.

*Question: SOD 1, 2 or total? Total

Changed as suggested by the esteem reviewer.

*Ll227/229-230: please write the abbreviations in full when used for the first time.

Changed as suggested by the esteem reviewer.  

*Figure1: please use arrows and letters on the figures to indicate the alpha and beta cells, as well as other structures of interest. Scale should just be a bar and the value be written down in the caption of the figure. Can the images have a slightly higher resolution? It would be interesting to  have an opinion as to why there is a change in the shape and size of the Langerhans islets.

Done as directed, and the images were replaced by others with higher resolution. 

In the rat islets are about 200 µm in diameter. Islets are approximately spherical to ovoid in shape and random sections exhibit a range of diameters. The largest islets are found in the tail of the rat pancreas, the part closest to the spleen (Elayat et al., 1995). The average numbers of cells are varied in different areas in the rat pancreas such as lower duodenal, upper duodenal, gastric and splenic areas (Elayat et al., 1995).

  • Elayat, Ahmed A., MOSTAFA M. el-Naggar, and Mohammad Tahir. "An immunocytochemical and morphometric study of the rat pancreatic islets." Journal of anatomy 186, no. Pt 3 (1995): 629.

*Figure 2 and 3 and 4: same. Please indicate structures with arrows and label accordingly.

Changed as suggested by the esteem reviewer.

*Please separate the graphics as new figures, increase their size and resolution. They are unreadable.

Improvement were made to the figures to make them easier to read.

*Please use SEM instead of SE.

Changed as suggested by the esteem reviewer.

*Figure 5: Please increase the resolution of the graphic, remove the excessive framing lines.

Changed as suggested by the esteem reviewer.

Section 3 - Discussion

*Please try to find some synonyms to ameliorate, it is rather repetitive.

Changed as suggested by the esteem reviewer.

*L331: In addition (caps)

Changed as suggested by the esteem reviewer.

*L332: concomitant administration of GLB, AJE or a combination of both in STZ…

Changed as suggested by the esteem reviewer.

*LL347-348: have not returned to normal values

Changed as suggested by the esteem reviewer.

*LL348-349: stimulation, instead of encouragement

Changed as suggested by the esteem reviewer.

*L374: Please make sure there is consistency in American and British English

Changed as suggested by the esteem reviewer.

*LL377-381: you compare one study by Aggarwal et al 2015 with another study… of Aggarwal et al, 2015

Wang et al., 2011

Wang ZQ, Ribnicky D, Zhang XH, Zuberi A, Raskin I, Yu Y, Cefalu WT. An extract of Artemisia dracunculus L. enhances insulin receptor signaling and modulates gene expression in skeletal muscle in KK-Ay mice. The Journal of nutritional biochemistry. 2011 Jan 1;22(1):71-8.

*L379: in pancreatic islets, not beta cells in pancreatic cells

Changed as suggested by the esteem reviewer.

*L384: AJ?

Changed as suggested by the esteem reviewer.

*L385: ( instead of ((

Changed as suggested by the esteem reviewer.

*L454: may be attributed.

Changed as suggested by the esteem reviewer.

Reviewer 2 Report

The article is very interesting and with very important results for the control of the glucose in the diabetic models. Nevertheless there are details with the writing that I suggest must be reviewed:

In the Abstract and Simple summary is not mentioned the concentration of the Glyburide.

To review the term: Pancreases or pancreas? Line 25, 48 and Fig 1.

Line 74:  (Qi et 74 al., 2010; and Ghorbani, 2014).

Line 182-190, 559-563 and Table 4. Effect on total hemoglobin and glycosylated hemoglobin levels, the time or period of treatment is not indicated: 2, 4, 8 weeks? 

Line 199-216 and Table 5. Effect on serum lipid profile, the time or period of treatment is not indicated: 2, 4, 8 weeks? 

Line 331: (60 mg/kg). In addition, the…

fig. or Fig.  

Author Response

Comments and Suggestions for Authors

The article is very interesting and with very important results for the control of the glucose in the diabetic models. Nevertheless there are details with the writing that I suggest must be reviewed:

*In the Abstract and Simple summary is not mentioned the concentration of the Glyburide.

Changed as suggested by the esteem reviewer.

*To review the term: Pancreases or pancreas? Line 25, 48 and Fig 1.

Changed as suggested by the esteem reviewer.

*Line 74:  (Qi et 74 al., 2010; and Ghorbani, 2014).

Changed as suggested by the esteem reviewer.

*Line 182-190, 559-563 and Table 4. Effect on total hemoglobin and glycosylated hemoglobin levels, the time or period of treatment is not indicated: 2, 4, 8 weeks? 

Changed as suggested by the esteem reviewer.

*Line 199-216 and Table 5. Effect on serum lipid profile, the time or period of treatment is not indicated: 2, 4, 8 weeks? 

Changed as suggested by the esteem reviewer.

*Line 331: (60 mg/kg). In addition, the…

Changed as suggested by the esteem reviewer.

*fig. or Fig.  

Changed as suggested by the esteem reviewer.

Reviewer 3 Report

This is an interesting topic and the research has potential for the therapeutics for diabetics.

However, the study is mostly descriptive. Also, the authors have used the Artemisia judaica extract. The authors discussed about possible active components in the Artemisia judaica extract. It is helpful that the authors can isolate some active components in the extract and test them as well.

Author Response

Comments and Suggestions for Authors

This is an interesting topic and the research has potential for the therapeutics for diabetics.

*However, the study is mostly descriptive. Also, the authors have used the Artemisia judaica extract. The authors discussed about possible active components in the Artemisia judaica extract. It is helpful that the authors can isolate some active components in the extract and test them as well.

We are in the process of isolation and identification of vulgarin from the extract and test in the same manner. However, isolating the compound in enough amount is time consuming and require many resources. Hope that we will have an article using the pure isolate in near future.

Reviewer 4 Report

The manuscript is written clearly and well-structured and presents a significant amount of experimental work. Some minor comments are as below.

  1. Line 94-97 Any example (references) for these drug-herb interactions?
  2. Abbreviations must be defined in tables and figures so that they are intelligible without reference to the text.
  3. Fig 2-4, Why “n=7”?
  4. Fig 5 What is “*”?
  5. Line 539-540 Rephrase this sentence, as group I was healthy rats and the other 6 groups were diabetic rats.
  6. Line 551 Correct the typo “2ed”.
  7. Clarify which statistical analysis was used, “Dunnett’s multiple comparison tests” (line 630) or “Tukey’s multiple comparison post hoc test” (each figure legend).

Author Response

The manuscript is written clearly and well-structured and presents a significant amount of experimental work. Some minor comments are as below.

*Line 94-97 Any example (references) for these drug-herb interactions?

Required information were added to the introduction.

*Abbreviations must be defined in tables and figures so that they are intelligible without reference to the text.

Done

*Fig 2-4, Why “n=7”?

7 microscopic high-power fields

*Fig 5 What is “*”?

Changed as suggested by the esteem reviewer.

*Line 539-540 Rephrase this sentence, as group I was healthy rats and the other 6 groups were diabetic rats.

Changed as suggested by the esteem reviewer.

*Line 551 Correct the typo “2ed”.

Changed as suggested by the esteem reviewer.

*Clarify which statistical analysis was used, “Dunnett’s multiple comparison tests” (line 630) or “Tukey’s multiple comparison post hoc test” (each figure legend).

Clarified.

Round 2

Reviewer 3 Report

The authors need to discuss the limitations of this study. 

Author Response

The authors need to discuss the limitations of this study. 

At the end of the discussion section a paragraph was added describing the limitations of the current study and the future plan to cover some of these limitations. Changes are marked in blue colour.